# Hamiltonian Neural Networks

**Sam Greydanus**
Google Brain
sgrey@google.com

**Misko Dzamba**
PetCube
mouse9911@gmail.com

**Jason Yosinski**
Uber AI Labs
yosinski@uber.com

## Abstract

Even though neural networks enjoy widespread use, they still struggle to learn the basic laws of physics. How might we endow them with better inductive biases? In this paper, we draw inspiration from Hamiltonian mechanics to train models that learn and respect exact conservation laws in an unsupervised manner. We evaluate our models on problems where conservation of energy is important, including the two-body problem and pixel observations of a pendulum. Our model trains faster and generalizes better than a regular neural network. An interesting side effect is that our model is perfectly reversible in time.

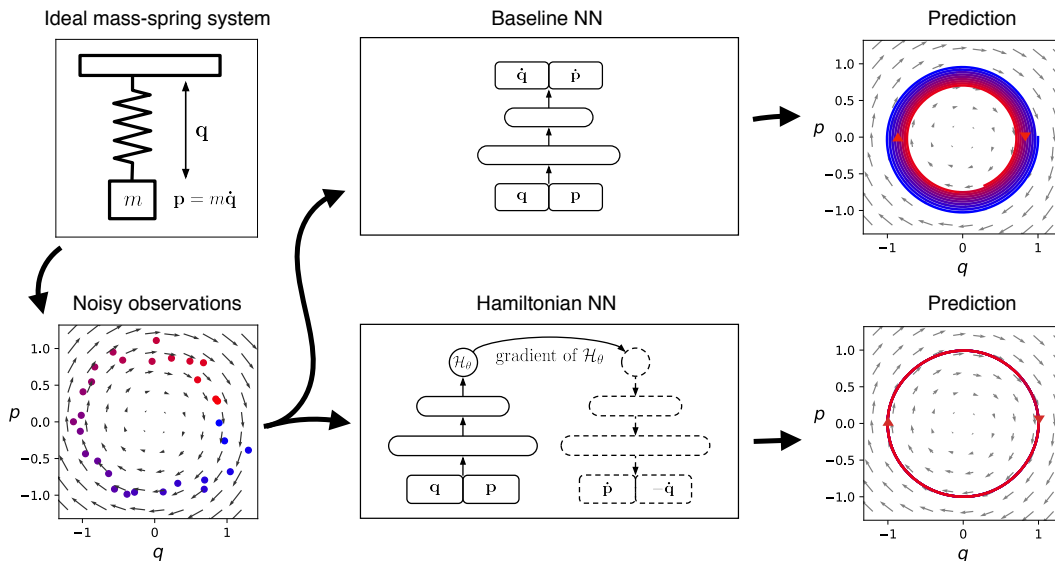

Figure 1: Learning the Hamiltonian of a mass-spring system. The variables $q$ and $p$ correspond to position and momentum coordinates. As there is no friction, the baseline's inner spiral is due to model errors. By comparison, the Hamiltonian Neural Network learns to *exactly* conserve a quantity that is analogous to total energy.

## 1 Introduction

Neural networks have a remarkable ability to learn and generalize from data. This lets them excel at tasks such as image classification [21], reinforcement learning [45, 26, 37], and robotic dexterity [1, 22]. Even though these tasks are diverse, they all share the same underlying physical laws. For example, a notion of gravity is important for reasoning about objects in an image, training an RL agent to walk, or directing a robot to manipulate objects. Based on this observation, researchers have become increasingly interested in finding physics priors that transfer across tasks [43, 34, 17, 10, 6, 40].

Untrained neural networks do not have physics priors; they learn approximate physics knowledge directly from data. This generally prevents them from learning *exact* physical laws. Consider the frictionless mass-spring system shown in Figure 1. Here the total energy of the system is being conserved. More specifically, this particular system conserves a quantity proportional to $q^2 + p^2$, where $q$ is the position and $p$ is the momentum of the mass. The baseline neural network in Figure 1 learns an approximation of this conservation law, and yet the approximation is imperfect enough that a forward simulation of the system drifts over time to higher or lower energy states. Can we define a class of neural networks that will precisely conserve energy-like quantities over time?

In this paper, we draw inspiration from Hamiltonian mechanics, a branch of physics concerned with conservation laws and invariances, to define *Hamiltonian Neural Networks*, or *HNNs*. We begin with an equation called the Hamiltonian, which relates the state of a system to some conserved quantity (usually energy) and lets us simulate how the system changes with time. Physicists generally use domain-specific knowledge to find this equation, but here we try a different approach:

> *Instead of crafting the Hamiltonian by hand, we propose parameterizing it with a neural network and then learning it directly from data.*

Since almost all physical laws can be expressed as conservation laws, our approach is quite general [27]. In practice, our model trains quickly and generalizes well[1]. Figure 1, for example, shows the outcome of training an HNN on the same mass-spring system. Unlike the baseline model, it learns to conserve an energy-like quantity.

## 2 Theory

**Predicting dynamics.** The hallmark of a good physics model is its ability to predict changes in a system over time. This is the challenge we now turn to. In particular, our goal is to learn the dynamics of a system using a neural network. The simplest way of doing this is by predicting the next state of a system given the current one. A variety of previous works have taken this path and produced excellent results [41, 14, 43, 34, 17, 6]. There are, however, a few problems with this approach.

The first problem is its notion of discrete "time steps" that connect neighboring states. Since time is actually continuous, a better approach would be to express dynamics as a set of differential equations and then integrate them from an initial state at $t_0$ to a final state at $t_1$. Equation 1 shows how this might be done, letting $\mathbf{S}$ denote the time derivatives of the coordinates of the system[2]. This approach has been under-explored so far, but techniques like Neural ODEs take a step in the right direction [7].

$$(\mathbf{q}_1, \mathbf{p}_1) \;=\; (\mathbf{q}_0, \mathbf{p}_0) \;+\; \int_{t_0}^{t_1} \mathbf{S}(\mathbf{q}, \mathbf{p}) \; dt \tag{1}$$

The second problem with existing methods is that they tend not to learn exact conservation laws or invariant quantities. This often causes them to drift away from the true dynamics of the system as small errors accumulate. The HNN model that we propose ameliorates both of these problems. To see how it does this — and to situate our work in the proper context — we first briefly review Hamiltonian mechanics.

**Hamiltonian Mechanics.** William Hamilton introduced Hamiltonian mechanics in the 19[th] century as a mathematical reformulation of classical mechanics. Its original purpose was to express classical mechanics in a more unified and general manner. Over time, though, scientists have applied it to nearly every area of physics from thermodynamics to quantum field theory [29, 32, 39].

In Hamiltonian mechanics, we begin with a set of coordinates $(\mathbf{q}, \mathbf{p})$. Usually, $\mathbf{q} = (q_1, ..., q_N)$ represents the positions of a set of objects whereas $\mathbf{p} = (p_1, ..., p_N)$ denotes their momentum. Note how this gives us $N$ coordinate pairs $(q_1, p_1)...(q_N, p_N)$. Taken together, they offer a complete description of the system. Next, we define a scalar function, $\mathcal{H}(\mathbf{q}, \mathbf{p})$ called the Hamiltonian so that

$$\frac{d\mathbf{q}}{dt} \;=\; \frac{\partial \mathcal{H}}{\partial \mathbf{p}}, \quad \frac{d\mathbf{p}}{dt} \;=\; -\frac{\partial \mathcal{H}}{\partial \mathbf{q}} \; . \tag{2}$$

Equation 2 tells us that moving coordinates in the direction $\mathbf{S}_\mathcal{H} = \left(\frac{\partial \mathcal{H}}{\partial \mathbf{p}}, -\frac{\partial \mathcal{H}}{\partial \mathbf{q}}\right)$ gives us the time evolution of the system. We can think of $\mathbf{S}$ as a vector field over the inputs of $\mathcal{H}$. In fact, it is a special kind of vector field called a "symplectic gradient". Whereas moving in the direction of the gradient of $\mathcal{H}$ changes the output as quickly as possible, moving in the direction of the symplectic gradient *keeps the output exactly constant*. Hamilton used this mathematical framework to relate the position and momentum vectors $(\mathbf{q}, \mathbf{p})$ of a system to its total energy $E_{tot} = \mathcal{H}(\mathbf{q}, \mathbf{p})$. Then, he found $\mathbf{S}_\mathcal{H}$ using Equation 2 and obtained the dynamics of the system by integrating this field according to Equation 1. This is a powerful approach because it works for almost any system where the total energy is conserved.

Hamiltonian mechanics, like Newtonian mechanics, can predict the motion of a mass-spring system or a single pendulum. But its true strengths only become apparent when we tackle systems with many degrees of freedom. Celestial mechanics, which are chaotic for more than two bodies, are a good example. A few other examples include many-body quantum systems, fluid simulations, and condensed matter physics [29, 32, 39, 33, 9, 12].

**Hamiltonian Neural Networks.** In this paper, we propose learning a parametric function for $\mathcal{H}$ instead of $\mathbf{S}_\mathcal{H}$. In doing so, we endow our model with the ability to learn *exactly* conserved quantities from data in an unsupervised manner. During the forward pass, it consumes a set of coordinates and outputs a single scalar "energy-like" value. Then, before computing the loss, we take an in-graph gradient of the output with respect to the input coordinates (Figure A.1). It is with respect to this gradient that we compute and optimize an $L_2$ loss (Equation 3).

$$\mathcal{L}_{HNN} = \left\| \frac{\partial \mathcal{H}_\theta}{\partial \mathbf{p}} - \frac{\partial \mathbf{q}}{\partial t} \right\|_2 + \left\| \frac{\partial \mathcal{H}_\theta}{\partial \mathbf{q}} + \frac{\partial \mathbf{p}}{\partial t} \right\|_2 \tag{3}$$

For a visual comparison between this approach and the baseline, refer to Figure 1 or Figure 1(b). This training procedure allows HNNs to learn conserved quantities analogous to total energy straight from data. Apart from conservation laws, HNNs have several other interesting and potentially useful properties. First, they are perfectly reversible in that the mapping from $(\mathbf{q}, \mathbf{p})$ at one time to $(\mathbf{q}, \mathbf{p})$ at another time is bijective. Second, we can manipulate the HNN-conserved quantity (analogous to total energy) by integrating along the gradient of $\mathcal{H}$, giving us an interesting counterfactual tool (e.g. "What would happen if we added 1 Joule of energy?"). We'll discuss these properties later in Section 6.

## 3   Learning a Hamiltonian from Data

Optimizing the gradients of a neural network is a rare approach. There are a few previous works which do this [42, 35, 28], but their scope and implementation details diverge from this work and from one another. With this in mind, our first step was to investigate the empirical properties of HNNs on three simple physics tasks.

**Task 1: Ideal Mass-Spring.** Our first task was to model the dynamics of the frictionless mass-spring system shown in Figure 1. The system's Hamiltonian is given in Equation 4 where $k$ is the spring constant and $m$ is the mass constant. For simplicity, we set $k = m = 1$. Then we sampled initial coordinates with total energies uniformly distributed between $[0.2, 1]$. We constructed training and test sets of 25 trajectories each and added Gaussian noise with standard deviation $\sigma^2 = 0.1$ to every data point. Each trajectory had 30 observations; each observation was a concatenation of $(\mathbf{q}, \mathbf{p})$.

$$\mathcal{H} = \frac{1}{2}kq^2 + \frac{p^2}{2m} \tag{4}$$

**Task 2: Ideal Pendulum.** Our second task was to model a frictionless pendulum. Pendulums are nonlinear oscillators so they present a slightly more difficult problem. Writing the gravitational constant as $g$ and the length of the pendulum as $l$, the general Hamiltonian is

$$\mathcal{H} = 2mgl(1 - \cos q) + \frac{l^2 p^2}{2m} \tag{5}$$

Once again we set $m = l = 1$ for simplicity. This time, we set $g = 3$ and sampled initial coordinates with total energies in the range $[1.3, 2.3]$. We chose these numbers in order to situate the dataset along the system's transition from linear to nonlinear dynamics. As with Task 1, we constructed training and test sets of 25 trajectories each and added the same amount of noise.

**Task 3: Real Pendulum.** Our third task featured the position and momentum readings from a real pendulum. We used data from a *Science* paper by Schmidt & Lipson [35] which also tackled the problem of learning conservation laws from data. This dataset was noisier than the synthetic ones and it did not *strictly* obey any conservation laws since the real pendulum had a small amount of friction. Our goal here was to examine how HNNs fared on noisy and biased real-world data.

## 3.1 Methods

In all three tasks, we trained our models with a learning rate of $10^{-3}$ and used the Adam optimizer [20]. Since the training sets were small, we set the batch size to be the total number of examples. On each dataset we trained two fully-connected neural networks: the first was a baseline model that, given a vector input $(\mathbf{q}, \mathbf{p})$ output the vector $(\partial \mathbf{q}/\partial t, \partial \mathbf{p}/\partial t)$ directly. The second was an HNN that estimated the same vector using the derivative of a scalar quantity as shown in Equation 2 (also see Figure A.1). Where possible, we used analytic time derivatives as the targets. Otherwise, we calculated finite difference approximations. All of our models had three layers, 200 hidden units, and `tanh` activations. We trained them for 2000 gradient steps and evaluated them on the test set.

We logged three metrics: $L_2$ train loss, $L_2$ test loss, and mean squared error (MSE) between the true and predicted total energies. To determine the energy metric, we integrated our models according to Equation 1 starting from a random test point. Then we used MSE to measure how much a given model's dynamics diverged from the ground truth. Intuitively, the loss metrics measure our model's ability to fit individual data points while the energy metric measures its stability and conservation of energy over long timespans. To obtain dynamics, we integrated our models with the fourth-order Runge-Kutta integrator in `scipy.integrate.solve_ivp` and set the error tolerance to $10^{-9}$ [30].

## 3.2 Results

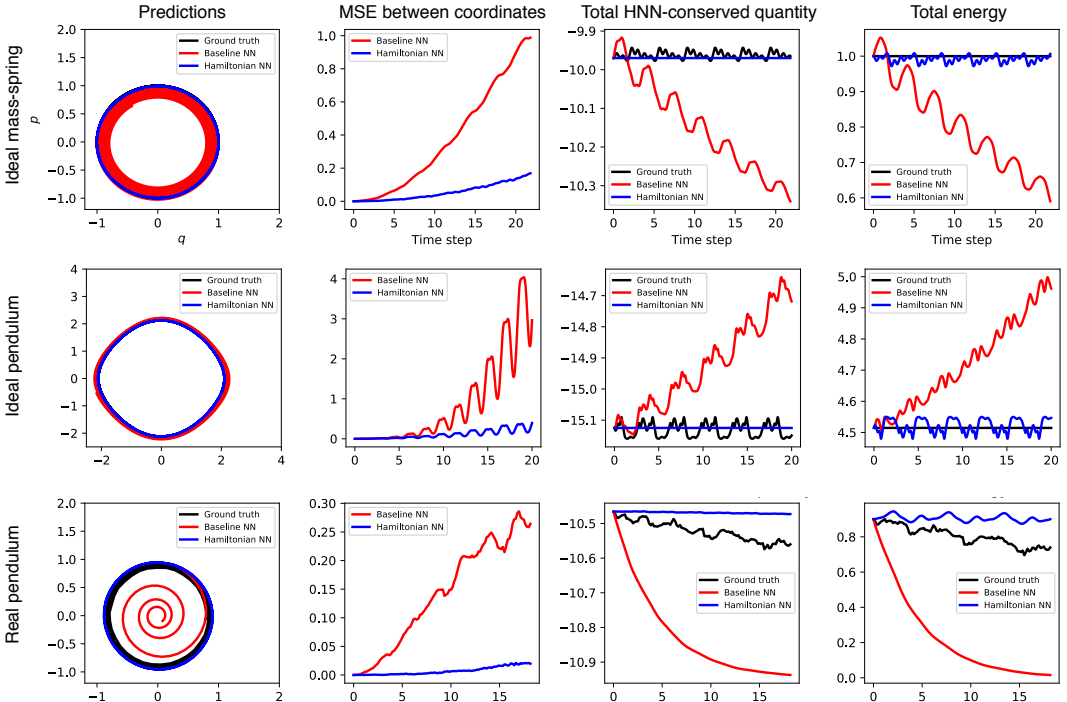

Figure 2: Analysis of models trained on three simple physics tasks. In the first column, we observe that the baseline model's dynamics gradually drift away from the ground truth. The HNN retains a high degree of accuracy, even obscuring the black baseline in the first two plots. In the second column, the baseline's coordinate MSE error rapidly diverges whereas the HNN's does not. In the third column, we plot the quantity conserved by the HNN. Notice that it closely resembles the total energy of the system, which we plot in the fourth column. In consequence, the HNN roughly conserves total energy whereas the baseline does not.

We found that HNNs train as quickly as baseline models and converge to similar final losses. Table 1 shows their relative performance over the three tasks. But even as HNNs tied with the baseline on on loss, they dramatically outperformed it on the MSE energy metric. Figure 2 shows why this is the case: as we integrate the two models over time, various errors accumulate in the baseline and it eventually diverges. Meanwhile, the HNN conserves a quantity that closely resembles total energy and diverges more slowly or not at all.

It's worth noting that the quantity conserved by the HNN is not equivalent to the total energy; rather, it's something very close to the total energy. The third and fourth columns of Figure 2 provide a useful comparison between the HNN-conserved quantity and the total energy. Looking closely at the spacing of the $y$ axes, one can see that the HNN-conserved quantity has the same scale as total energy, but differs by a constant factor. Since energy is a relative quantity, this is perfectly acceptable[3].

The total energy plot for the real pendulum shows another interesting pattern. Whereas the ground truth data does not quite conserve total energy, the HNN roughly conserves this quantity. This, in fact, is a fundamental limitation of HNNs: they assume a conserved quantity exists and thus are unable to account for things that violate this assumption, such as friction. In order to account for friction, we would need to model it separately from the HNN.

## 4 Modeling Larger Systems

Having established baselines on a few simple tasks, our next step was to tackle a larger system involving more than one pair of $(p, q)$ coordinates. One well-studied problem that fits this description is the two-body problem, which requires four $(p, q)$ pairs.

$$\mathcal{H} = \frac{|\mathbf{p_{CM}}|^2}{m_1 + m_2} + \frac{|\mathbf{p_1}|^2 + |\mathbf{p_2}|^2}{2\mu} + g\frac{m_1 m_2}{|\mathbf{q_1} - \mathbf{q_2}|^2} \tag{6}$$

**Task 4: Two-body problem.** In the two-body problem, point particles interact with one another via an attractive force such as gravity. Once again, we let $g$ be the gravitational constant and $m$ represent mass. Equation 6 gives the Hamiltonian of the system where $\mu$ is the reduced mass and $\mathbf{p_{CM}}$ is the momentum of the center of mass. As in previous tasks, we set $m_1 = m_2 = g = 1$ for simplicity. Furthermore, we restricted our experiments to systems where the momentum of the center of mass was zero. Even so, with eight degrees of freedom (given by the $x$ and $y$ position and momentum coordinates of the two bodies) this system represented an interesting challenge.

### 4.1 Methods

Our first step was to generate a dataset of 1000 near-circular, two-body trajectories. We initialized every trajectory with center of mass zero, total momentum zero, and radius $r = \|\mathbf{q_2} - \mathbf{q_1}\|$ in the range $[0.5, 1.5]$. In order to control the level of numerical stability, we chose initial velocities that gave perfectly circular orbits and then added Gaussian noise to them. We found that scaling this noise by a factor of $\sigma^2 = 0.05$ produced trajectories with a good balance between stability and diversity.

We used fourth-order Runge-Kutta integration to find 200 trajectories of 50 observations each and then performed an 80/20% train/test set split over trajectories. Our models and training procedure were identical to those described in Section 3 except this time we trained for 10,000 gradient steps and used a batch size of 200.

### 4.2 Results

The HNN model scaled well to this system. The first row of Figure 3 suggests that it learned to conserve a quantity nearly equal to the total energy of the system whereas the baseline model did not.

The second row of Figure 3 gives a qualitative comparison of trajectories. After one orbit, the baseline dynamics have completely diverged from the ground truth whereas the HNN dynamics have only accumulated a small amount of error. As we continue to integrate up to $t = 50$ and beyond (Figure B.1), both models diverge but the HNN does so at a much slower rate. Even as the HNN

diverges from the ground truth orbit, its total energy remains stable rather than decaying to zero or spiraling to infinity. We report quantitative results for this task in Table 1. Both train and test losses of the HNN model were about an order of magnitude lower than those of the baseline. The HNN did a better job of conserving total energy, with an energy MSE that was several orders of magnitude below the baseline.

Having achieved success on the two-body problem, we ran the same set of experiments on the chaotic three-body problem. We show preliminary results in Appendix B where once again the HNN outperforms its baseline by a considerable margin. We opted to focus on the two-body results here because the three-body results still need improvement.

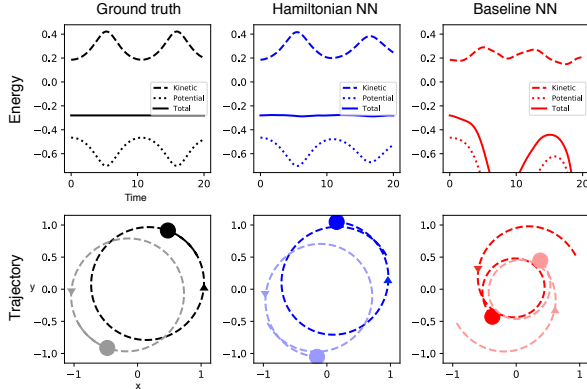

Figure 3: Analysis of an example 2-body trajectory. The dynamics of the baseline model do not conserve total energy and quickly diverge from ground truth. The HNN, meanwhile, approximately conserves total energy and accrues a small amount of error after one full orbit.

# 5 Learning
# a Hamiltonian from Pixels

One of the key strengths of neural networks is that they can learn abstract representations directly from high-dimensional data such as pixels or words. Having trained HNN models on position and momentum coordinates, we were eager to see whether we could train them on arbitrary coordinates like the latent vectors of an autoencoder.

**Task 5: Pixel Pendulum.** With this in mind, we constructed a dataset of pixel observations of a pendulum and then combined an autoencoder with an HNN to model its dynamics. To our knowledge this is the first instance of a Hamiltonian learned directly from pixel data.

## 5.1 Methods

In recent years, OpenAI Gym has been widely adopted by the machine learning community as a means for training and evaluating reinforcement learning agents [5]. Some works have even trained world models on these environments [15, 16]. Seeing these efforts as related and complimentary to our work, we used OpenAI Gym's `Pendulum-v0` environment in this experiment.

First, we generated 200 trajectories of 100 frames each[4]. We required that the maximum absolute displacement of the pendulum arm be $\frac{\pi}{6}$ radians. Starting from 400 x 400 x 3 RGB pixel observations, we cropped, desaturated, and downsampled them to 28 x 28 x 1 frames and concatenated each frame with its successor so that the input to our model was a tensor of shape `batch` x 28 x 28 x 2. We used two frames so that velocity would be observable from the input. Without the ability to observe velocity, an autoencoder without recurrence would be unable to ascertain the system's full state space.

In designing the autoencoder portion of the model, our main objective was simplicity and trainability. We chose to use fully-connected layers in lieu of convolutional layers because they are simpler. Furthermore, convolutional layers sometimes struggle to extract even simple position information [23]. Both the encoder and decoder were composed of four fully-connected layers with `relu` activations and residual connections. We used 200 hidden units on all layers except the latent vector **z**, where we used two units. As for the HNN component of this model, we used the same architecture and parameters as described in Section 3. Unless otherwise specified, we used the same training procedure as described in Section 4.1. We found that using a small amount of weight decay, $10^{-5}$ in this case, was beneficial.

**Losses.** The most notable difference between this experiment and the others was the loss function. This loss function was composed of three terms: the first being the HNN loss, the second being a classic autoencoder loss ($L_2$ loss over pixels), and the third being an auxiliary loss on the autoencoder's

latent space:

$$\mathcal{L}_{CC} = \left\| \mathbf{z}_{\mathbf{p}}^t - (\mathbf{z}_{\mathbf{q}}^t - z_{\mathbf{q}}^{t+1}) \right\|_2 \tag{7}$$

The purpose of the auxiliary loss term, given in Equation 7, was to make the second half of $\mathbf{z}$, which we'll label $\mathbf{z_p}$, resemble the derivatives of the first half of $\mathbf{z}$, which we'll label $\mathbf{z_q}$. This loss encouraged the latent vector $(\mathbf{z_q}, \mathbf{z_p})$ to have roughly same properties as canonical coordinates $(\mathbf{q}, \mathbf{p})$. These properties, measured by the Poisson bracket relations, are necessary for writing a Hamiltonian. We found that the auxiliary loss did not degrade the autoencoder's performance. Furthermore, it is not domain-specific and can be used with any autoencoder with an even-sized latent space.

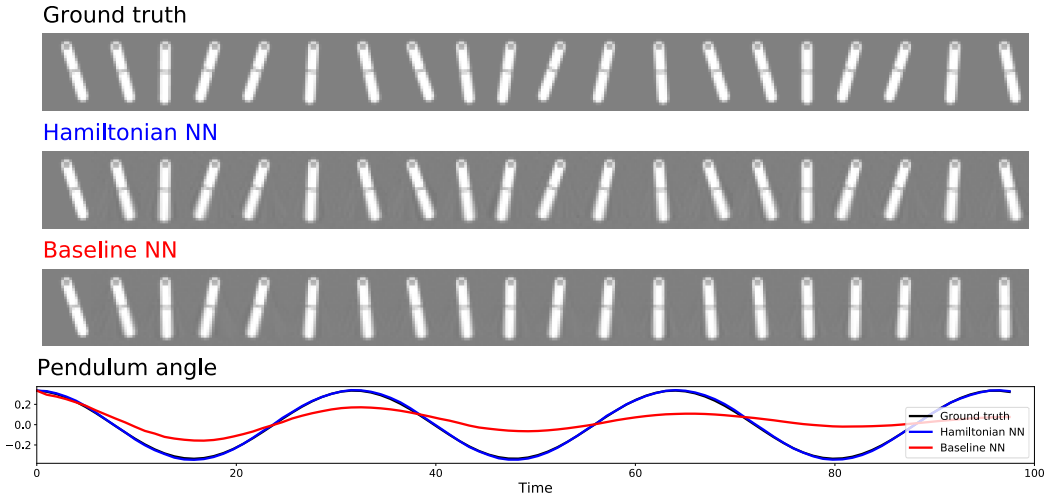

Figure 4: Predicting the dynamics of the pixel pendulum. We train an HNN and its baseline to predict dynamics in the latent space of an autoencoder. Then we project to pixel space for visualization. The baseline model rapidly decays to lower energy states whereas the HNN remains close to ground truth even after hundreds of frames. It mostly obscures the ground truth line in the bottom plot.

## 5.2 Results

Unlike the baseline model, the HNN learned to conserve a scalar quantity analogous to the total energy of the system. This enabled it to predict accurate dynamics for the system over much longer timespans. Figure 4 shows a qualitative comparison of trajectories predicted by the two models. As in previous experiments, we computed these dynamics using Equation 2 and a fourth-order Runge-Kutta integrator. Unlike previous experiments, we performed this integration in the latent space of the autoencoder. Then, after integration, we projected to pixel space using the decoder network. The HNN and its baseline reached comparable train and test losses, but once again, the HNN dramatically outperformed the baseline on the energy metric (Table 1).

Table 1: Quantitative results across all five tasks. Whereas the HNN is competitive with the baseline on train/test loss, it dramatically outperforms the baseline on the energy metric. All values are multiplied by $10^3$ unless noted otherwise. See Appendix A for a note on train/test split for Task 3.

|  | Train loss | | Test loss | | Energy | |
|---|---|---|---|---|---|---|
| Task | Baseline | HNN | Baseline | HNN | Baseline | HNN |
| 1: Ideal mass-spring | $37 \pm 2$ | $37 \pm 2$ | $37 \pm 2$ | $\mathbf{36 \pm 2}$ | $170 \pm 20$ | $\mathbf{.38 \pm .1}$ |
| 2: Ideal pendulum | $33 \pm 2$ | $33 \pm 2$ | $\mathbf{35 \pm 2}$ | $36 \pm 2$ | $42 \pm 10$ | $\mathbf{25 \pm 5}$ |
| 3: Real pendulum | $2.7 \pm .2$ | $9.2 \pm .5$ | $\mathbf{2.2 \pm .3}$ | $6.0 \pm .6$ | $390 \pm 7$ | $\mathbf{14 \pm 5}$ |
| 4: Two body ($\times 10^6$) | $33 \pm 1$ | $3.0 \pm .1$ | $30 \pm .1$ | $\mathbf{2.8 \pm .1}$ | $6.3e4 \pm 3e4$ | $\mathbf{39 \pm 5}$ |
| 5: Pixel pendulum | $18 \pm .2$ | $19 \pm .2$ | $\mathbf{17 \pm .3}$ | $18 \pm .3$ | $9.3 \pm 1$ | $\mathbf{.15 \pm .01}$ |

# 6 Useful properties of HNNs

While the main purpose of HNNs is to endow neural networks with better physics priors, in this section we ask what other useful properties these models might have.

**Adding and removing energy.** So far, we have seen that integrating the symplectic gradient of the Hamiltonian can give us the time evolution of a system but we have not tried following the Riemann gradient $\mathbf{R}_{\mathcal{H}} = \left( \frac{\partial \mathcal{H}}{\partial \mathbf{q}}, \frac{\partial \mathcal{H}}{\partial \mathbf{p}} \right)$. Intuitively, this corresponds to adding or removing some of the HNN-conserved quantity from the system. It's especially interesting to alternate between integrating $\mathbf{R}_{\mathcal{H}}$ and $\mathbf{S}_{\mathcal{H}}$. Figure 5 shows how we can take advantage of this effect to "bump" the pendulum to a higher energy level. We could imagine using this technique to answer counterfactual questions e.g. "What would have happened if we applied a torque?"

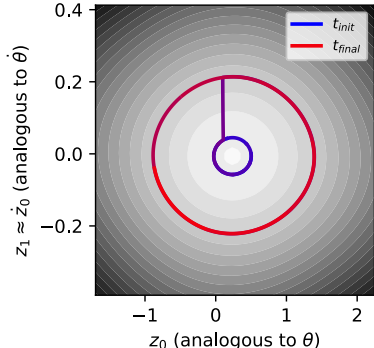

Figure 5: Visualizing integration in the latent space of the Pixel Pendulum model. We alternately integrate $\mathbf{S}_{\mathcal{H}}$ at low energy (blue circle), $\mathbf{R}_{\mathcal{H}}$ (purple line), and then $\mathbf{S}_{\mathcal{H}}$ at higher energy (red circle).

**Perfect reversibility.** As neural networks have grown in size, the memory consumption of transient activations, the intermediate activations saved for backpropagation, has become a notable bottleneck. Several works propose semi-reversible models that construct one layer's activations from the activations of the next [13, 25, 19]. Neural ODEs also have this property [7]. Many of these models are only approximately reversible: their mappings are not quite bijective. Unlike those methods, our approach is guaranteed to produce trajectories that are perfectly reversible through time. We can simply refer to a result from Hamiltonian mechanics called Liouville's Theorem: *the density of particles in phase space is constant*. What this implies is that any mapping $(\mathbf{q}_0, \mathbf{p}_0) \rightarrow (\mathbf{q}_1, \mathbf{p}_1)$ is bijective/invertible.

# 7 Related work

**Learning physical laws from data.** Schmidt & Lipson [35] used a genetic algorithm to search a space of mathematical functions for conservation laws and recovered the Lagrangians and Hamiltonians of several real systems. We were inspired by their approach, but used a neural neural network to avoid constraining our search to a set of hand-picked functions. Two recent works are similar to this paper in that the authors sought to uncover physical laws from data using neural networks [18, 4]. Unlike our work, they did not explicitly parameterize Hamiltonians.

**Physics priors for neural networks.** A wealth of previous works have sought to furnish neural networks with better physics priors. Many of these works are domain-specific: the authors used domain knowledge about molecular dynamics [31, 38, 8, 28], quantum mechanics [36], or robotics [24] to help their models train faster or generalize. Others, such as Interaction Networks or Relational Networks were meant to be fully general [43, 34, 2]. Here, we also aimed to keep our approach fully general while introducing a strong and theoretically-motivated prior.

**Modeling energy surfaces.** Physicists, particularly those studying molecular dynamics, have seen success using neural networks to model energy surfaces [3, 11, 36, 44]. In particular, several works have shown dramatic computation speedups compared to density functional theory [31, 38, 8]. Molecular dynamics researchers integrate the derivatives of energy in order to obtain dynamics, just as we did in this work. A key difference between these approaches and our own is that 1) we emphasize the Hamiltonian formalism 2) we optimize the gradients of our model (though some works do optimize the gradients of a molecular dynamics model [42, 28]).

# 8 Discussion

Whereas Hamiltonian mechanics is an old and well-established theory, the science of deep learning is still in its infancy. Whereas Hamiltonian mechanics describes the real world from first principles, deep learning does so starting from data. We believe that Hamiltonian Neural Networks, and models like them, represent a promising way of bringing together the strengths of both approaches.

# 9 Acknowledgements

Sam Greydanus would like to thank the Google AI Residency Program for providing extraordinary mentorship and resources. The authors would like to thank Nic Ford, Trevor Gale, Rapha Gontijo Lopes, Keren Gu, Ben Caine, Mark Woodward, Stephan Hoyer, Jascha Sohl-Dickstein, and many others for insightful conversations and support.

Special thanks to James and Judy Greydanus for their feedback and support from beginning to end.

## Footnotes

[1]We make our code available at `github.com/greydanus/hamiltonian-nn`.

[2]Any coordinates that describe the state of the system. Later we will use position and momentum $(\mathbf{p}, \mathbf{q})$.

[3]To see why energy is relative, imagine a cat that is at an elevation of 0 m in one reference frame and 1 m in another. Its potential energy (and total energy) will differ by a constant factor depending on frame of reference.

[4]Choosing the "no torque" action at every timestep.

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
