[Supplementary Material]



# A   Supplementary Information for Tasks 1-3

(a) Baseline NN

(b) Hamiltonian NN

Figure 6: HNN schema. The forward pass of an HNN is composed of a forward pass through a differentiable model as well as a backpropagation step through the model.

**Training details.** We selected hyperparameters using a coarse grid search over learning rates $\{10^{-1}, 10^{-2}, 10^{-3}\}$, layer widths $\{100, 200, 300\}$, activations $\{\texttt{tanh}, \texttt{relu}\}$, and batch size where relevant $\{100, 200\}$. The main objective of this work was not to produce state-of-the-art results, so the settings we chose were aimed simply at producing models that gave good qualitative performance on the tasks at hand. We used weight decay of $10^{-4}$ on the first three tasks.

We trained all of these experiments on a desktop CPU.

**The large test loss on Task 3.** The test losses we report on Task 3 are singificantly larger than the training losses. This discrepancy is a result of the way we partitioned the training and test sets. The dataset provided by [29] consisted of just a single trajectory from a real pendulum, as shown in the second panel of Figure 7(c). Also, we needed to evaluate our model's performance over a series of adjacent time steps in order to measure the energy MSE metric. For this reason, we were forced to use the first $4/5$ of this trajectory for training (black vectors in Figure 7(c)) and the last $1/5$ for evaluation (red vectors).

The consequence of this train/test split is that our test set had a slightly different distribution from our training set, producing larger test losses compared to train losses. We found that the relative magnitudes of the test losses between the baseline and HNN models were informative, which is why we report them. We did not perform this ungainly train/test split on the other two tasks in this section.

(a) Task 1: Ideal mass-spring

(b) Task 2: Ideal pendulum

(c) Task 3: Real pendulum

Figure 7: More qualitative results comparing the HNN to a baseline neural network on the first three physics tasks. From top to bottom: Task 1: Ideal mass-spring, Task 2: ideal pendulum, Task 3: Real pendulum.

# B    Supplementary Information for Task 4: Two-body problem

**Training details.** We selected hyperparameters with a grid search as described in the previous section. Again, the main objective of this work was not to produce state-of-the-art results, so the settings we chose were aimed simply at producing models that gave good qualitative performance on the tasks at hand. We did not use weight decay on this task, though when we tried a weight decay of $10^{-4}$ or results did not change significantly.

We trained this experiment on a desktop CPU.

(a) Baseline NN

(b) Hamiltonian NN

Figure 8: More qualitative results for the orbit task. Numerical errors accumulate in the baseline model until the bodies end up traveling in opposite directions. The total energy diverges towards infinity as well. In comparison, the HNN's trajectory diverges from the ground truth but continues to roughly conserve the total energy of the system.

Figure 9: Comparison of how well the HNN conserves total energy compared to the baseline its baseline on the two-body task.

**Three body problem.** As mentioned briefly in the body of the paper, we also trained our models on the three body problem. The results we report here show a relative advantage to using the HNN over the baseline model. However, both models struggled to accurately model the dynamics of the three-body problem, which is why we relegated these results to the Appendix. Going forward, we hope to improve these results to the point where they can play a more substantial role in Section 4.

411 Table 2 gives a summary of quantitative results and Figure 10 shows a qualitative analysis of the
412 models we trained on this task.

Table 2: Quantitative results for the three-body problem.

|  | Train loss | | Test loss | | Energy MSE | |
|---|---|---|---|---|---|---|
|  | Baseline | HNN | Baseline | HNN | Baseline | HNN |
| Task 4b: 3-body problem | 0.096 | 0.080 | **0.380** | 0.488 | 103.9 | **0.039** |

Figure 10: Analysis of an example three-body trajectory. The baseline model does not conserve total energy and quickly diverges from ground truth. The HNN, meanwhile, roughly conserves total energy and its trajectories resemble the ground truth.

## C   Supplementary Information for Task 5: Pixel Pendulum

**Training details.** We selected hyperparameters with a grid search as described in the previous section. We used a weight decay of $10^{-5}$ on this experiment. We found that, unlike previous experiments, weight decay had a significant impact on results. We suspect that this is because the scale of the gradients on the weights of the HNN portion of the model were different from the scale of the gradients of the weights of the autoencoder portion of the model.

We trained this experiment on a desktop CPU.

(a) Latent space of the autoencoder

(b) Contour plot of HNN-conserved quantity in latent space

Figure 11: Latent space plots from the Pixel Pendulum model. Note that the learned latent space bears a strong resemblance to the true phase space of a pendulum. In particular, there is a faint diamond shape to the outer contour lines of Figure 11(b). This pattern is reminiscent of the nonlinear dynamics we observed in the ideal pendulum phase space plot of Figure 2 (row 2, column 1)