[Reviews · NeurIPS 2019]

Reviewer 1



As I mentioned, up to my knowledge, the idea of using Hamiltonian equations as loss functions of NNs is new, interesting and easy to follow. However I am not convinced that it can be applied to a large set of physical problems. The major draw back is that the Hamiltonian equations should be known in advance by the designers of the model rather than learned from data. Another short-coming is their trivial experimental results. As a matter of fact, I do not find much point in the presented toy tasks 1, 2 and even 3, as the maximum information that the network is potentially able to learn is to estimate the noise parameter as otherwise the provided prior knowledge is sufficient to solve these tasks (and therefore no neural net is needed). The last task is much more interesting because the NN learns to link the raw data (pixels) into the quantities for which the Hamiltonian is defined. However even this task is in a sense too simple and does not convince me that such an approach can be applied to any real world problem.

Reviewer 2



This paper is very well written, nicely motivated and introduces a general principle to design neural network for data with conservation laws using Hamiltonian mechanics. Contrary to what the authors state, including energy conservation into neural networks and optimizing its gradients is now common procedure in this domain, for example: - Pukrittayakamee et al. Journal of Chemical Physics, 130(13). 2009 - Behler. Physical Chemistry Chemical Physics, 13(40). 2011 - Gastegger. Journal of Chemical Theory and Computation, 11(5), 2187-2198. 2015 - Schuett et al, NeurIPS 30 / Journal of Chemical Physics 148(24). 2017 - Yao et al., Chemical science 9(8). 2018 The proposed approach constitutes a generalization of neural networks for high-dimensional potential energy surface by including the momentum in the input to arrive at the Hamiltonian formalism. For classical systems, as presented in this paper, it seems that this addition is rather counter-productive: while the change of momentum is described by the potential (see references above), the change of positions directly follows from the equations of motion and does not require an additional derivative of the network. This is both more computationally efficient and generalizes by design to all initial momenta (provided the corresponding positions stay close to the training manifold). On the other hand, I am not convinced that the proposed architecture would still work when applying a trained model to a different energy level. The mentioned property of reversibility is not convincing as it is not described how this property can help to make back-propagation more efficient. Reversing the trajectory only allows to discard the intermediate time step, which are already present in the training set. For the network itself, one still needs to use the full back-propagation of each time step to get the gradient w.r.t. the parameters. Beyond that, the mentioned reversibility is given for all predictions that are by-design conservative force fields, such as the earlier mentioned neural network potentials. A strong point of the paper is the application to the pixel pendulum. This could inspire new and creative applications of neural networks that encode conservation laws. I would still guess, that similar results can be obtained from a pure potential approach without momentum inputs. It would be interesting to see future, more general applications, where this is no longer possible. In the author feedback, my main issue with the paper has not been addressed sufficiently. The Hamiltonian formalism is more general than predicting just the potential energy + derivatives (which amounts to energy conservation from classical mechanic), but this is not taken advantage of. The author's response, that potential energy approaches need potential energy labels is not correct: they can also be trained solely on derivatives (e.g., see JCP ref above). A concrete example: A consequence of the more general Hamiltonian formalism is that the network would need to be retrained for a pendulum with a different total energy since the impulse at the same pendulum position would be different. In contrast, a network using potential energy derivatives + equations of motions can handle different energy levels, since it does not depend on the impulse and takes only positions. Apart from this, the approach is a valuable contribution with great potential in other areas than classical mechanics.

Reviewer 3



1. Originality: To the best of my knowledge, modeling hamiltonian of a dynamic system using NN is novel. Though there are concurrent works with the similar theme, this paper is, from my point of view, the most clear and thorough one among them. The related works are well cited. 2. Quality This paper is technically sound. This work is self-contained and did a good job to prove the concept it introduces. The evaluation is thorough, yet a bit simple, but is powerful enough to prove the concept of this paper. 3. Clarity: This paper is well written. It give a gentle introduction to hamiltonian mechanics for readers who may not have the proper background. 4. Significance: This work provides a novel, concrete and practical methodology for learning dynamic systems in a physically grounded fashion. Moreover, this modeling strategy has great potential since it may not rely on the exact coordinate frame which the system is defined, as hamiltonians can be defined on any valid generalized frame which suit the constraint. This direction definitely requires more thoughts and efforts and should be of significance to the community.

[Author Response · NeurIPS 2019]

We thank the reviewers for both their favorable and their critical feedback. We have used the latter to improve and clarify the paper. Below we group and respond to key issues, highlighting the changes we've made:

**1. Tasks 1-3 are perhaps too simple (R1).** We agree that tasks 1-3 are simple in that they are easy to solve with existing methods. We included them not to showcase the real-world potential of HNNs but simply to allow for a scientific investigation of their basic properties. Task 1 was used to show that optimizing the symplectic gradient of a neural network is a viable and stable strategy. We then asked whether an HNN could learn a nonlinear vector field, and Task 2 provided a sanity check that it could. We then asked whether an HNN would learn something sensible from real-world data, and Task 3 showed that it did. Once these sanity checks were complete, we applied our approach to the more complex and exciting Tasks 4 and 5 which we believe better showcase the power of HNNs. While we could remove tasks 1-3 due to their simplicity, we feel the rigor and clarity are improved thanks to their inclusion.

**2. Hamiltonian is not generally proportional to $q^2 + p^2$ (line 19) (R1).** This part was poorly written: we had not intended to claim that the Hamiltonian will be generally proportional to $q^2 + p^2$. We've updated the text as shown in $a$ in the figure below.

**3. Fix issues with structure: Table 1 is before tasks 4 and 5; Figure 6 not found (R1).** We agree that these parts were a mess. Table 1 has been moved to page 7 of the paper so that the reader can view it after reading the explanations of Tasks 4 and 5. Table 1 was poorly formatted; we have added confidence intervals and updated it to be much more readable. These changes are reflected in $c$ in the figure below. As for Figure 6, it has been moved to the Appendix and is now more clearly referenced as such as shown in $b$ in the figure below.

**4. Add references for optimizing gradients of neural networks (R2)** Thanks for the five references. All are great additions to the paper and we have added them along with a discussion of the overlap (see part $d$ in the figure below).

**5. Show a simple control example, especially with the latent coordinate space (R3).** Great idea! We believe that applying HNNs to control tasks will produce compelling results. However, to limit the scope of this paper to that of introducing the core concept of an HNN, we decided to leave this application to future work.

**6. Model systems with (a) friction/damping or (b) contact modeling (R3).** Thanks for these suggestions. When we were writing the paper we discussed some related experiments: how to model (a) a damped harmonic oscillator and (b) balls bouncing in a box. Ultimately, we decided to focus this paper on a thorough and principled investigation of the key effect and leave these more complicated scenarios to future work.

**7. Outline applications where the equations of motion are not sufficient/where HNNs are needed (R2). Address significance of HNNs, especially for solving real-world problems (R1).** One key difference between HNNs and potential energy-based approaches is that the latter generally require a reference energy obtained via electronic structure calculations (Equation 10 in Behler 20ll and Equations 1,2 in Pukrittayakamee 2009). Meanwhile, HNNs are trained in an unsupervised manner: we do not require reference energies. Because of this, HNNs are promising for datasets that have unusual coordinate systems (e.g. Task 5: Pixel Pendulum) such that the reference energies are not trivial to compute. We are very excited about real-world applications of this technique. Since releasing a preprint of this paper, we have heard from one group that is currently running experiments with HNNs to calibrate quantum computers and another group that is considering using HNNs to learn/calibrate the gaits of a biped robot.



[Meta-Review · NeurIPS 2019]

This paper proposes a novel approach for using neural networks to model physical systems via the Hamiltonian. Reviewers agreed this was correct, novel, and useful.